# On the relationship between Normalising Flows and Variational- and Denoising Autoencoders

**Alexey A. Gritsenko[2], Jasper Snoek[1] & Tim Salimans[1]**
[1] Google Brain, [2] Google AI Resident
{agritsenko,salimans,jsnoek}@google.com

## Abstract

Normalising Flows (NFs) are a class of likelihood-based generative models that have recently gained popularity. They are based on the idea of transforming a simple density into that of the data. We seek to better understand this class of models, and how they compare to previously proposed techniques for generative modeling and unsupervised representation learning. For this purpose we reinterpret NFs in the framework of Variational Autoencoders (VAEs), and present a new form of VAE that generalises normalising flows. The new generalised model also reveals a close connection to denoising autoencoders, and we therefore call our model the Variational Denoising Autoencoder (VDAE). Using our unified model, we systematically examine the model space between flows, variational autoencoders, and denoising autoencoders, in a set of preliminary experiments on the MNIST handwritten digits. The experiments shed light on the modeling assumptions implicit in these models, and they suggest multiple new directions for future research in this space.

## 1 Introduction and Background

Unsupervised learning offers the promise of leveraging unlabeled data to learn representations useful for downstream tasks when labeled data is scarce (van den Oord et al., 2017b), or even to generate novel data in domains where it is costly to obtain (Ha & Schmidhuber, 2018). Generative models are particularly appealing for this as they provide a statistical model of the data, $\mathbf{x}$, usually in the form of a joint probability density $p(\mathbf{x})$. The model's density function, its samples and representations can then be leveraged in applications ranging from semi-supervised learning (Kingma et al., 2014) and speech and (conditional) image synthesis (van den Oord et al., 2016a; Ledig et al., 2017; Guadarrama et al., 2017; Kingma & Dhariwal, 2018) to gene expression analysis (Grønbech et al., 2019) and molecule design (Gómez-Bombarelli et al., 2018).

In practice, data $\mathbf{x}$ is often high-dimensional and the optimization associated with learning $p(\mathbf{x})$ can be challenging due to an abundance of local minima (Sønderby et al., 2016) and difficulty in sampling from rich high-dimensional distributions (Murray & Salakhutdinov, 2009). Despite this, generative modelling has undergone a surge of advancements with recent developments in likelihood-based models (Kingma & Welling, 2013; Rezende et al., 2014; Dinh et al., 2014; van den Oord et al., 2016a) and Generative Adversarial Networks (GANs; Goodfellow et al. (2014)). The former class is particularly attractive, as it offers (approximate) likelihood evaluation and the ability to train models using likelihood maximisation, as well as interpretable latent representations.

Autoencoders have a rich history in the unsupervised learning literature owing to their intuitive and simple construction for learning complex latent representations of data. Through fitting a parameterised mapping from the data through a lower dimensional or otherwise constrained layer back to the same data, the model learns to summarise the data in a compact latent representation. Many variants of autoencoders have been proposed to encourage the model to better encode the underlying structure of the data though regularising or otherwise constraining the model (e.g., Rifai et al., 2011; Alain & Bengio, 2013; Arpit et al., 2016).

**Denoising Autoencoders (DAEs)** are a variant of the autoencoder under which noise is added to the input data that the model must then output noise-free, i.e. $\mathbf{x} = \mathbf{f}_\theta(\mathbf{x} + \boldsymbol{\epsilon})$ where $\boldsymbol{\epsilon}$ is sampled from a, possibly structured Vincent et al. (2008; 2010), noise distribution $\boldsymbol{\epsilon} \sim q(\boldsymbol{\epsilon})$. They are inspired by the idea that a good representation $\mathbf{z}$ would be robust to noise corrupting the data $\mathbf{x}$ and that adding noise would discourage the model from simply learning the identity mapping. Although DAEs have been cast as generative models (Bengio et al., 2013), sampling and computing likelihoods under the model remains challenging.

**Variational Autoencoders (VAEs)** instead assume a probabilistic latent variable model, in which $n$-dimensional data $\mathbf{x}$ correspond to $m$-dimensional latent representations $\mathbf{z}$ following some tractable prior distribution, i.e. $\mathbf{x} \sim p_\phi(\mathbf{x}|\mathbf{z})$ with $\mathbf{z} \sim p(\mathbf{z})$ (Kingma & Welling, 2013). The task is then to learn parameters $\phi$, which requires maximising the log marginal likelihood

$$\log p(\mathbf{x}) = \log \int p_\phi(\mathbf{x}|\mathbf{z}) \, p(\mathbf{z}) \, d\mathbf{z}. \tag{1}$$

In the majority of practical cases (e.g. $p_\phi(\mathbf{x}|\mathbf{z})$ taken to be a flexible neural network-conditional distribution) the above integral is intractable. A variational lower bound on the marginal likelihood is constructed using a variational approximation $q_\theta(\mathbf{z}|\mathbf{x})$ to the unknown posterior $p(\mathbf{z}|\mathbf{x})$:

$$\log p(\mathbf{x}) \geq \mathbb{E}_{q_\theta(\mathbf{z}|\mathbf{x})}[\log p_\phi(\mathbf{x}|\mathbf{z})] - \mathbb{KL}[q_\theta(\mathbf{z}|\mathbf{x})\,||\,p(\mathbf{z})] \tag{2}$$

The right-hand side of (2), denoted $\mathcal{L}(\theta, \phi)$, is known as the evidence lower bound (ELBO). It can be jointly optimised with stochastic optimisation w.r.t. parameters $\theta$ and $\phi$ in place of (1).

Conditionals $q_\theta(\mathbf{z}|\mathbf{x})$ and $p_\phi(\mathbf{x}|\mathbf{z})$ can be viewed respectively as probabilistically encoding data $\mathbf{x}$ in the latent space, and reconstructing it from samples of this encoding. The first term of the ELBO encourages good reconstructions, whereas the second term encourages the model's latent variables to be distributed according to the prior $p(\mathbf{z})$. Generating new data using this model is accomplished by reconstructing samples from the prior.

**Normalising Flows (NFs)** suppose that the sought distribution $p(\mathbf{x})$ can be obtained by warping a simple base density $p(\mathbf{z})$, e.g. a normal distribution (Rezende & Mohamed, 2015). They make use of the change of variables formula to obtain $p(\mathbf{x})$ through a learned invertible transformation $\mathbf{z} = \mathbf{f}_\theta(\mathbf{x})$ as

$$\log p_\theta(\mathbf{x}) = \log p(\mathbf{z}) + \log \left| \det \left( \frac{\partial \mathbf{f}_\theta(\mathbf{x})}{\partial \mathbf{x}} \right) \right|. \tag{3}$$

Typically, $\mathbf{f}_\theta : \mathbb{R}^n \to \mathbb{R}^n$ is obtained by stacking several simpler mappings, i.e. $\mathbf{f}_\theta = \mathbf{f}_\theta^1 \circ \ldots \circ \mathbf{f}_\theta^L$ and the log-determinant obtained as the sum of log-determinants of these mappings.

This formulation allows for *exact* maximum likelihood learning, but requires $\mathbf{f}_\theta$ to be invertible and to have a tractable inverse and Jacobian determinant. This restricts the flexibility of known transformations that can be used in NFs (Dinh et al., 2014; 2016; Behrmann et al., 2018) and leads to large and computationally intensive models in practice (Kingma & Dhariwal, 2018).

NFs can also be thought of as VAEs with encoder and decoder modelled as Dirac deltas $p_\theta(\mathbf{x}|\mathbf{z}) = \delta(\mathbf{f}_\theta(\mathbf{z}))$ and $q_\theta(\mathbf{z}|\mathbf{x}) = \delta(\mathbf{f}_\theta^{-1}(\mathbf{x}))$, constructed using a restricted set of transformations. Furthermore, because NFs model continuous density, to prevent trivial solutions with infinite point densities discrete data must be *dequantised* by adding random noise (Uria et al., 2013; Theis et al., 2015).

**The contribution** of this work is two-fold. First, we shed new light on the relationship between DAEs, VAEs and NFs, and discuss the pros and cons of these model classes. Then, we also introduce several extensions of these models, which we collectively refer to as the ***Variational Denoising Autoencoders (VDAEs)***.

In the most general form VDAEs generalise NFs and DAEs to discrete data and learned noise distributions. However, when the amount of injected noise is small, VDAE attains a form that allows for using non-invertible transformations (e.g. $\mathbf{f}_\theta : \mathbb{R}^n \to \mathbb{R}^m$, with $m \ll n$). We demonstrate

these theoretical advantages through preliminary experimental results on the binary and continuous versions of the MNIST dataset.

## 2   A VAE THAT GENERALISES NF AND DAE

We model data $\mathbf{x}$, a $n$ dimensional vector that can have either continuous or discrete support. As is customary for VAEs, our model for $\mathbf{x}$ is hierarchical and assumes a set of latent variables $\mathbf{z}$ with tractable prior distribution $p(\mathbf{z})$, and a flexible neural-network conditional distribution $p(\mathbf{x}|\mathbf{z})$. On top of this standard VAE setup, we specify the dimension of $\mathbf{z}$ to equal the dimension of the data $\mathbf{x}$. In order to form the variational lower bound to train this model, we need an approximate inference model, or *encoder*, $q_\theta\left(\mathbf{z}|\mathbf{x}\right)$. Here, we will use an encoder that samples the latents $\mathbf{z}$ as

$$\boldsymbol{\epsilon} \sim q\left(\boldsymbol{\epsilon}\right), \quad \tilde{\mathbf{x}} = \mathbf{x} + \boldsymbol{\epsilon}, \quad \mathbf{z} = \mathbf{f}_\theta(\tilde{\mathbf{x}}), \tag{4}$$

where $q\left(\boldsymbol{\epsilon}\right)$ is a tractable noise distribution and $\mathbf{f}_\theta(\tilde{\mathbf{x}})$ is a one-to-one transformation with tractable Jacobian-determinant. In order to use the encoder $q_\theta\left(\mathbf{z}|\mathbf{x}\right)$ implied by this procedure, we not only need to sample from it, but we must also evaluate its entropy for the KL-term in (2). To do this we make use of the fact that $\mathbf{z}$ is a one-to-one transformation of the noise $\boldsymbol{\epsilon}$, given the training data $\mathbf{x}$. Using the standard formulas for a change of variables, we thus get the following expression for the entropy of $q_\theta\left(\mathbf{z}|\mathbf{x}\right)$:

$$\mathbb{H}\left[q_\theta\left(\mathbf{z}|\mathbf{x}\right)\right] = \mathbb{H}\left[q\left(\boldsymbol{\epsilon}\right)\right] + \mathbb{E}_{\boldsymbol{\epsilon}} \log\left|\det\left(\frac{\partial \mathbf{f}_\theta\left(\mathbf{x}+\boldsymbol{\epsilon}\right)}{\partial \boldsymbol{\epsilon}}\right)\right| = \mathbb{H}\left[q\left(\boldsymbol{\epsilon}\right)\right] + \mathbb{E}_{q(\tilde{\mathbf{x}}|\mathbf{x})} \log\left|\det\left(\frac{\partial \mathbf{f}_\theta\left(\tilde{\mathbf{x}}\right)}{\partial \tilde{\mathbf{x}}}\right)\right|,$$

where $q\left(\tilde{\mathbf{x}}|\mathbf{x}\right)$ is a distribution whose sampling process is described in (4). Our variational lower bound (2) on the data log marginal likelihood then becomes

$$\log p(\mathbf{x}) \geq \mathbb{H}\left[q\left(\boldsymbol{\epsilon}\right)\right] + \mathbb{E}_{\boldsymbol{\epsilon}}\left[\log p(\mathbf{z}) + \log p(\mathbf{x}|\mathbf{z}) + \log\left|\det\left(\frac{\partial \mathbf{f}_\theta\left(\tilde{\mathbf{x}}\right)}{\partial \tilde{\mathbf{x}}}\right)\right|\right], \tag{5}$$

where again $\tilde{\mathbf{x}} = \mathbf{x} + \boldsymbol{\epsilon}$ and $\mathbf{z} = \mathbf{f}_\theta(\tilde{\mathbf{x}})$.

This is **similar to a denoising autoencoder** in that we try to reconstruct the original data $\mathbf{x}$ from the corrupted data $\tilde{\mathbf{x}}$ through the conditional model $p(\mathbf{x}|\mathbf{z})$. The difference with classical denoising autoencoders is that our objective has additional terms that regularise our latent representations $\mathbf{z}$ to be distributed according to a prior distribution $p(\mathbf{z})$. In addition, the proposed setup allows us to learn the noise distribution $q(\boldsymbol{\epsilon})$, where this is treated as a fixed hyperparameter in the literature on denoising autoencoders.

This model is also a **generalisation of normalising flows**. Specifically, consider the special case where we take

$$\begin{aligned} q\left(\boldsymbol{\epsilon}\right) &= \mathcal{N}\left(\mathbf{0}, \sigma^2 \mathbf{I}_n\right) \\ p(\mathbf{x}|\mathbf{z}) &= \mathcal{N}\left(\mathbf{f}_\theta^{-1}(\mathbf{z}), \sigma^2 \mathbf{I}_n\right) \\ \sigma^2 &\to 0 \end{aligned} \tag{6}$$

then the lower bound in (5) becomes the standard normalising flow log-likelihood (3). We provide a detailed derivation in Appendix A.

The advantage of our generalised model over standard normalising flows is that our model allows for non-zero noise level $\sigma^2$. Interestingly, successful applications of normalising flows in the literature often already add a significant amount of noise in order to dequantise the data, and empirical results suggest higher amounts of noise lead to models that produce better-looking samples (e.g. Kingma & Dhariwal (2018) model only the top $5$ bits of the data).

In addition, our model does not require tying the parameters of the encoder and decoder. Although we are still using a flow-based encoder $q_\theta(\mathbf{z}|\mathbf{x})$, our decoder is not restricted to a specific functional form. The conditional distribution $p(\mathbf{x}|\mathbf{z})$ can e.g. have discrete support if the data $\mathbf{x}$ is discrete, making our model naturally applicable to data such as text or other highly-structured data, without requiring an explicit dequantisation step. When adding a significant amount of noise, a decoupled decoder will generally be able to achieve a higher variational lower bound compared to using the tied-parameter decoder (6).

## 3 GENERALISING TO NON-INVERTIBLE ENCODERS

The VAE we proposed in Section 2 is more general than NFs, but it still requires an invertible one-to-one encoder with tractable Jacobian-determinant. This restricts our modeling choices since all transformations used in the encoder can only be chosen from a small set of transformations for which we know how to compute inverses and Jacobian-determinants. Additionally, the representation given by our encoder will be of the same dimension as our data $\mathbf{x}$, which may not be optimal for all applications (e.g. model-based reinforcement learning (Ha & Schmidhuber, 2018; Hafner et al., 2018) or compression (Ballé et al., 2018)). To relax these restrictions further we generalise our model to allow non-invertible encoders as well.

We proceed by taking our model from Section 2, with $\tilde{\mathbf{x}} = \mathbf{x} + \boldsymbol{\epsilon}$ and $\mathbf{z} = \mathbf{f}_\theta(\tilde{\mathbf{x}})$, and performing a Taylor expansion of the resulting latent variables $\mathbf{z}(\mathbf{x}, \boldsymbol{\epsilon})$ around $\boldsymbol{\epsilon} = \mathbf{0}$ (see Appendix B). This gives

$$\mathbf{z} = \mathbf{f}_\theta(\mathbf{x}) + \mathbf{J}\boldsymbol{\epsilon} + \mathcal{O}(\boldsymbol{\epsilon}^2),$$

where $\mathbf{J} \equiv \frac{\partial \mathbf{f}_\theta(\mathbf{x})}{\partial \mathbf{x}}$ is the Jacobian of $\mathbf{f}_\theta$.

For small noise levels, as used in Section 2, the $\mathcal{O}(\epsilon^2)$ term becomes negligible. If the noise distribution is Gaussian, i.e. $q(\boldsymbol{\epsilon}) = \mathcal{N}(\mathbf{0}, \sigma^2 \mathbf{I}_n)$, this means that for small $\sigma$ we get

$$q_\theta(\mathbf{z}|\mathbf{x}) = \mathcal{N}\left(\mathbf{f}_\theta(\mathbf{x}), \sigma^2 \mathbf{J}\mathbf{J}^T\right). \tag{7}$$

Using this form of encoder $q_\theta(\mathbf{z}|\mathbf{x})$, together with general prior $p(\mathbf{z})$ and conditional distribution $p(\mathbf{x}|\mathbf{z})$, we get a VAE that still generalises NFs but now also allows us to choose non-invertible non-one-to-one transformations $\mathbf{f}_\theta$. We refer to this even broader class of VAE as L-VDAE, for *Linearised*-VDAE.

### 3.1 LOG-DETERMINANT COMPUTATION

Evaluating the entropy $\mathbb{H}[q_\theta(\mathbf{z}|\mathbf{x})]$ in this case requires computing the log-determinant of the covariance matrix $\mathbf{C} = \mathbf{J}\mathbf{J}^T$ for the data $\mathbf{x}$:

$$\mathbb{H}[q_\theta(\mathbf{z}|\mathbf{x})] = \frac{1}{2}[m \log 2\pi + m + \log \det \mathbf{C}], \tag{8}$$

where $m$ is the dimensionality of $\mathbf{z}$. When using transformations $\mathbf{f}_\theta$ without a tractable Jacobian (e.g. the general Residual Network (ResNet; He et al. (2016)) blocks), we explicitly evaluate $\mathbf{C}$ and compute $\log \det \mathbf{C} = \sum_i^m \log \lambda_i$, where the eigenvalues $\lambda_i$ are obtained using the eigenvalue decomposition $\mathbf{C} = \mathbf{Q}\mathbf{\Lambda}\mathbf{Q}^T$ with $\mathbf{\Lambda} = \text{diag}(\lambda_i | i = 1, \ldots, m)$. The decomposition is further re-used in the backward pass when evaluating the derivative of the log-determinant using Jacobi's formula: $\frac{d}{d\mathbf{C}} \log \det \mathbf{C} = \mathbf{Q}\mathbf{\Lambda}^{-1}\mathbf{Q}^T$.

Evaluation of the Jacobian $\mathbf{J}$ can be done by performing reverse-mode automatic differentiation with respect to each element of $z$, thus incurs a factor of $m$ additional computational cost. Covariance matrix $\mathbf{C}$ is obtained using a single matrix multiplication and takes $\mathcal{O}(m^2 n)$ operations with the eigenvalue decomposition taking another $\mathcal{O}(m^3)$ operations.

Taken together, evaluation of (8) takes $\mathcal{O}(m^2 n)$ operations, which is comparable to the $\mathcal{O}(d^3)$ cost of Glow's `1x1` invertible convolutions in later layers (i.e. after repeating the use of the multi-scale architecture from Dinh et al. (2016) that trades spatial dimensions for channels), where $d$ refers to

the number of channels used in the convolution. This computational cost is permissive for small latent space dimensionalities $m$. However, scaling up L-VDAE to larger latent spaces would require stochastic approximations of the log-determinant (Han et al., 2015; 2018). These approximations can be implemented efficiently through Jacobian-vector and vector-Jacobian products, without evaluating $\mathbf{C}$ or $\mathbf{J}$ explicitly, and can be optimised directly by backpropagating through the approximation. With this approach computational complexity will be linear in $n$ subject to some regularity conditions.

## 3.2 Sampling from the variational posterior

Sampling from the Gaussian variational posterior $q_\phi$ is necessary for training and inference in L-VDAE. It can be accomplished using the standard reparameterisation trick (Kingma & Welling, 2013), where random normal noise $\boldsymbol{\omega} \sim \mathcal{N}(\mathbf{0}, \mathbf{I}_n)$ is transformed into the posterior sample as $\mathbf{z} = \mathbf{f}_\theta(\mathbf{x}) + \mathbf{J}\boldsymbol{\omega}$. We implement this as a Jacobian-vector product, which enables efficient sampling for cases when the Jacobian log-determinant of $\mathbf{f}_\theta$ is cheaper to evaluate than the Jacobian itself (e.g. when $\mathbf{f}_\theta$ is a flow).

## 4 Related work

VDAE blends ideas from the VAE and NFs literature and is closely related to both model families. It is most similar to methods that combined variational inference and NFs by using the latter as part of the approximate variational posterior (Rezende & Mohamed, 2015; Kingma et al., 2016b; van den Berg et al., 2018). These methods use a strategy in which samples from the (Gaussian) posterior are further transformed with a NF, whereas in VDAE the posterior distribution is implicitly defined using a sampling procedure inspired by DAE, where posterior samples are obtained by transforming data with added noise using NFs.

VDAE is a natural formulation of DAEs as probabilistic models. It is conceptually similar to the Denoising VAEs (Im et al., 2017), which propose an alternative probabilistic formulation of DAEs as VAEs. The method of Im et al., however, does not generalise NFs and, in contrast to VDAE, it requires explicitly choosing the type and amount of corruption noise.

The idea of challenging the default choice of using uniform noise for dequantisation in NFs was also explored in Flow++ (Ho et al., 2019), where the authors learned a flexible conditional noise model $q(\boldsymbol{\epsilon}|\mathbf{x})$ as NF itself. Our sampling procedure (4) is similar to dequantisation in Flow++, as it can be viewed as a result of applying a NF to dequantisation given by an implicitly conditioned noise model. The main differences, however, are that in VDAE the decoder reconstructs the original (quantised) data, which is also what makes our model applicable to highly-structured data; and, in contrast to Flow++, VDAE can inject substancially more noise than a single dequantisation bin.

In relation to VAEs, the linearised form of VDAE can be viewed as an extension of the vanilla VAE (Kingma & Welling, 2013) that replaces the diagonal Gaussian posterior with a Jacobian-based full covariance posterior. It is thus similar to methods that extend VAE with more flexible prior (e.g. autoregressive (Chen et al., 2016) or mixture (Tomczak & Welling, 2017)) or variational posterior (e.g. full covariance Gaussian (Kingma et al., 2016a) or mixture (Nalisnick et al., 2016; Miller et al., 2017)) distributions. Notably, unlike some of these methods, L-VDAE does not increase the number of parameters of the inference or generative networks.

As a method that increases flexibility of transformations in NFs, L-VDAE with non-invertible encoders can be compared to Invertible Residual Networks (i-ResNets; Behrmann et al. (2018)) and FFJORD (Grathwohl et al., 2018). These methods too depart from the requirement of restricting the form of the Jacobian of the resulting transformation. Both, i-ResNets and FFJORD also drop the requirement of having an *analytical* inverse, which is similar to how VDAE seeks to learn an approximate inverse using its decoder network. However, unlike VDAE, these methods guarantee invertibility and provide ways of computing the exact inverse. Notably, the methods differ considerably in how they achieve the above generalisations.

In i-ResNets Behrmann et al. make use of the ResNet network architecture (He et al., 2016) and identify conditions on the eigenvalues of the residual blocks, under which they parameterise invertible mappings. They then make use of spectral normalisation (Miyato et al., 2018) to guarantee that the condition is satisfied throughout training; and employ fixed point iteration to invert the residual blocks

Table 1: Model test performance comparison on **continuous** MNIST (in bits per dimension; lower is better).

| LATENTS | METHOD | NLL | -ELBO |
|---|---|---|---|
| 16 | VAE | 6.55 | 6.63 |
| | L-VDAE | **6.53** | **6.62** |
| 32 | VAE | 6.15 | 6.25 |
| | L-VDAE | **6.10** | **6.22** |
| 64 | VAE | 5.78 | 5.89 |
| | L-VDAE | **5.74** | **5.86** |
| 128 | VAE | 5.49 | 5.66 |
| | L-VDAE | **5.40** | **5.58** |
| $28 \times 28$ | L-VDAE [*] | 5.76 | **6.18** |
| | VDAE: RESNET | **5.74** | 6.19 |
| $28 \times 28$ | NICE [**] | 4.18 | - |
| | NICE | 4.36 | - |
| | REALNVP | 1.06 | - |
| | GLOW | 1.05 | - |
| | FFJORD | 0.99 | - |
| | I-RESNET | 1.06 | - |

[*] Using a NICE flow for the variational posterior.
[**] Results based on our implementation.

for generation. i-ResNets further lift the restriction on the form of the Jacobian in a computationally tractable way by using Taylor series expansion in conjunction with stochastic trace estimation (Hutchinson, 1990).

FFJORD (Grathwohl et al., 2018) is inspired by the re-interpretation of ResNets and NFs as discrete approximations of solutions to the initial value problem of some underlying ODE continuously transforming data $\mathbf{x}$ (from data-space to the latent $\mathbf{z}$-space; Chen et al. (2018)). Grathwohl et al. (2018); Chen et al. (2018) parameterise this ODE as a neural network $\mathbf{f}_\theta (\mathbf{z} (t), t)$ to obtain Continuous-time Normalising Flows (CNFs), in which the change in log-density at time $t$ is given by the instantaneous change of variables formula

$$\frac{\partial \log p_\theta (\mathbf{z} (t))}{\partial t} = - \mathrm{Tr} \left[ \frac{\partial \mathbf{f}_\theta (\mathbf{z} (t), t)}{\partial \mathbf{z} (t)} \right]. \tag{9}$$

The right-hand side of (9) is given by the trace of the Jacobian of transformation $\mathbf{f}_\theta$ instead of the log-determinant as in NFs. Combined with the use of stochastic trace estimation (Hutchinson, 1990), this difference alleviates the need to restrict transformations $\mathbf{f}_\theta$ to those with a tractable Jacobian log-determinant. However, the use of ODEs also necessitates employing an ODE solver to integrate (9) for every evaluation of, and backpropagation through $\log p_\theta (\mathbf{z} (t))$. The number of function evaluations required for this increases with training and may become prohibitively large (Grathwohl et al., 2018).

Finally, VDAE is loosely related to autoregressive generative models, as they both fall into the class of likelihood-based generative models. Autoregressive models factorise likelihood of high-dimensional data as a product of simple per-dimension conditional distributions, i.e. $p(\mathbf{x}) = \prod_i p(x_i | x_0, \ldots, x_{i-1})$ (van den Oord et al., 2016a;b). Factorised structure of these models necessitates sequential sampling, and a good choice of the ordering of dimensions of $\mathbf{x}$. Overcoming these challenges in practice often requires highly engineered solutions, for example as in van den Oord et al. (2017a) or Menick & Kalchbrenner (2018). Furthermore, data representations formed by hidden layers of autoregressive models appear to be more challenging to manipulate than in VAEs or NFs (Kingma & Dhariwal, 2018).

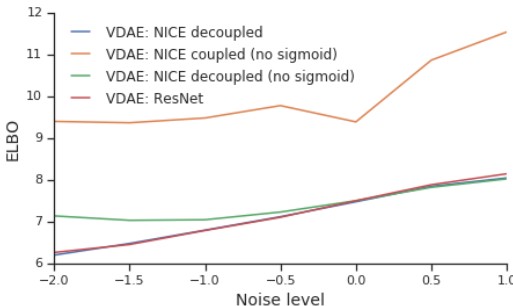

Figure 1: Test set negative ELBO in bits per dimension (lower is better) on **continuous** MNIST shown as a function of *fixed* noise level $\log \sigma$ for VDAE.

Table 2: Model test performance comparison on dynamic **binary** MNIST data (in nats; lower is better).

| LATENTS | METHOD | NLL | -ELBO |
|---|---|---|---|
| 16 | VAE | $83.46 \pm 0.18$ | $87.17 \pm 0.20$ |
| | L-VDAE | $\mathbf{82.68 \pm 0.14}$ | $\mathbf{86.07 \pm 0.15}$ |
| 32 | VAE | $82.96 \pm 0.16$ | $86.47 \pm 0.23$ |
| | L-VDAE | $\mathbf{81.54 \pm 0.24}$ | $\mathbf{84.81 \pm 0.24}$ |
| 64 | VAE | $82.77 \pm 0.20$ | $86.41 \pm 0.28$ |
| | L-VDAE | $\mathbf{81.49 \pm 0.21}$ | $\mathbf{84.92 \pm 0.25}$ |
| 128 | VAE | $82.99 \pm 0.19$ | $86.73 \pm 0.26$ |
| | L-VDAE | $\mathbf{81.82 \pm 0.18}$ | $\mathbf{85.39 \pm 0.23}$ |
| $28 \times 28$ | L-VDAE [*] | $96.47 \pm 0.23$ | $116.94 \pm 0.37$ |
| | VDAE: RESNET | $\mathbf{92.03 \pm 0.28}$ | $\mathbf{110.00 \pm 0.36}$ |

[*] Using a NICE flow for the variational posterior.

## 5 EXPERIMENTS

We performed empirical studies of the performance of VDAE on the image generation task on the MNIST dataset (LeCun, 1998), comparing it to a VAE implementation with a fully factorised Gaussian posterior and to the NICE (Dinh et al., 2014) normalising flow as baselines.

For the VDAE encoder we used additive couplings to construct $\mathbf{f}_\theta$ from the implicit variational posterior; and, unless otherwise specified, fully-connected ResNet blocks followed by a sigmoid transformation to obtain the decoder parameters $\mu_\phi$ and $p_\phi$. A Gaussian distribution $\mathcal{N}\left(\mu_\phi\left(\mathbf{z}\right), \lambda \mathbf{I}_n\right)$ with a learned parameter $\lambda$ was used for the continuous MNIST decoder; and $\text{Bernoulli}\left(p_\phi\left(\mathbf{z}\right)\right)$ for binary MNIST.

Similarly, unless otherwise specified, ResNet blocks with linear projection layers to change dimensionality were used for the L-VDAE encoder and decoder. Details of the chosen architectures can be found in Appendix D.

### 5.1 CONTINUOUS MNIST

To model discrete 8-bit pixel values with continuous density models, we followed the procedure of Uria et al. (2013) to dequantise the data, and added noise $u \sim \mathcal{U}\left(0, 1\right)$ to the pixel values prior to normalising them to $[0, 1]$. Note that for VDAE this was done prior and in addition to adding noise $\epsilon$ from the posterior sampling procedure (4) to the inputs.

**Decoupled encoder and decoder** We start by confirming for a range of noise levels and architectures that de-coupling the encoder and decoder networks in VDAE allows for achieving higher

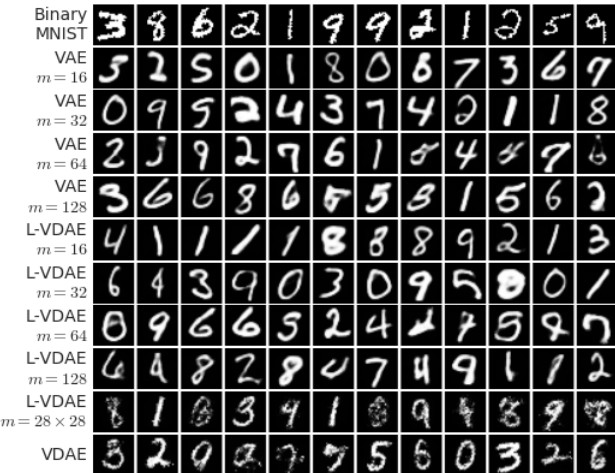

Figure 2: Samples from our models trained on the **binary** MNIST dataset; samples from the dataset are provided for reference (top row).

ELBOs. Figure 1 compares ELBO attained by VDAE with Gaussian noise $q(\epsilon) = \mathcal{N}(\mathbf{0}, \sigma^2 \mathbf{I})$ for a range of fixed $\sigma$. The results show that any decoupling of the weights improves over the coupled network, in which the NICE flow is used in the encoder and its inverse - in the decoder. Specifically, we observe that for architectures with a sigmoid activation in the last layer of the decoder, the ELBO rapidly improves with decreasing noise levels. Based on these results, in the following experiments we only consider the more general ResNet architecture in the VDAE decoder.

We report average test set performance over 10 training runs; when sufficiently large, standard deviations are also given. Qualitative samples are drawn from models with the best test ELBO among the training runs. NLL was estimated via 5000 importance samples as in Rezende et al. (2014).

**Quantitative results**  We now consider the cases when i) noise variance $\sigma^2$, in case of VDAE; or ii) in case of L-VDAE, the covariance scale $\sigma$ from (7); are optimised together with the model. Results of these experiments are shown in Table 1. For ease of presentation, we also include evaluation results for existing flow models (reproduced from Behrmann et al. (2018)).

We first note that for cases when the latent dimensionality is smaller than the input space (i.e. $m < n$), L-VDAE consistently outperforms the VAE baseline in terms of the achieved ELBO, albeit by a small margin. This is consistent with L-VDAE having a more powerful variational posterior. Moreover, for L-VDAE increasing the dimensionality of the latent space consistently improves the variational lower bound. Surprisingly, L-VDAE with $n = m$ and VDAE break this trend and do not improve on the ELBOs obtained for $m = 128$. We also note that neither of our proposed extensions manage to achieve likelihoods comparable to NFs, including the NICE baseline.

Both shortcomings could be explained by the difference in architectures between the methods. In contrast to the L-VDAE with $m = n$, which employs a NICE flow in the encoder, L-VDAE with $m < n$ makes use of the more expressive ResNet blocks. Similarly, the flexibility of the NICE flow used in VDAE for the implicit posterior may be insufficient for a denoising VAE. We also observe that when using a NICE flow in the decoder, VDAE outperforms L-VDAE in terms of likelihood, signalling that the VDAE approach can further improve on the linearised models, if combined with a more powerful flow.

**Qualitative results**  We found that without additional regularisation, such as fixing the decoder variance $\lambda^2$ or the noise variance $\sigma^2$ to values larger than what would have been learned by the model, or assigning a higher weight to the KL-term in the optimisation objective, our models would not produce high-quality samples for the continuous MNIST dataset. We thus omit **continuous** MNIST

model samples from the main text, but explore the effect of fixing the noise variance on sample quality in Appendix E.

## 5.2 BINARY MNIST

To explore the applicability of VDAE to structured data, we applied it to the binarised version of the MNIST dataset. As is customary for dynamic MNIST, digits were binarised by sampling from a Bernoulli distribution given by the pixel intensities. Results in Table 2 mirror those we observed on the continuous MNIST, namely L-VDAE consistently achieves higher ELBO than the VAE baseline, which tends to improve as the latent dimensionality grows; and L-VDAE and VDAE, which make use of NICE in the decoder, attain significantly worse likelihood despite the increased dimensionality. Finally, VDAE also improves on L-VDAE with a NICE encoder.

However, as shown in Figure 2, and in contrast to the continuous MNIST results, all our models produce plausible handwritten digit samples.

## 6 CONCLUSION AND FUTURE WORK

We introduced Variational Denoising Autoencoders (VDAEs), a family of models the bridges the gap between VAEs, NFs and DAEs. Our model extends NFs to discrete data and non-invertible encoders that use lower-dimensional latent representations. Preliminary experiments on the MNIST handwritten digits demonstrate that our model can be successfully applied to data with discrete support, attaining competitive likelihoods and generating plausible digit samples. We also identified a failure mode of our models, in which their performance does not scale well to cases when latent and input dimensionalities are the same (i.e. when a flow-based encoder is used).

Future work should address limitations of the method identified in our experiments. In particular, replacing additive coupling blocks with the more powerful invertible convolutions, affine coupling blocks and invertible residual blocks (Dinh et al., 2016; Kingma & Dhariwal, 2018; Behrmann et al., 2018) can significantly improve the variational posterior for high dimensions. It can also be interesting to explicitly condition the transformation $\mathbf{f}_\theta$ used for defining the posterior sampling procedure on the data $\mathbf{x}$, for example by defining $\mathbf{f}_\theta(\mathbf{x}, \boldsymbol{\epsilon}) \equiv \mathbf{f}_{\mathbf{x},\theta}(\boldsymbol{\epsilon})$ using a hyper-network (Ha et al., 2016).

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

## A  VDAE IS A GENERALISATION OF NORMALISING FLOWS

For convenience we start by repeating the variational lower bound for our VDAE model, as presented in Section 2:

$$\log p(\mathbf{x}) \geq \mathbb{H}[q(\boldsymbol{\epsilon})] + \mathbb{E}_{\boldsymbol{\epsilon}} \left[ \log p(\mathbf{z}) + \log p(\mathbf{x}|\mathbf{z}) + \log \left| \det \left( \frac{\partial \mathbf{f}_\theta(\tilde{\mathbf{x}})}{\partial \tilde{\mathbf{x}}} \right) \right| \right],$$

where $\boldsymbol{\epsilon} \sim q(\boldsymbol{\epsilon})$, $\tilde{\mathbf{x}} = \mathbf{x} + \boldsymbol{\epsilon}$ and $\mathbf{z} = \mathbf{f}_\theta(\tilde{\mathbf{x}})$.

We then examine the special case where we choose $q(\boldsymbol{\epsilon}) = \mathcal{N}\left(\mathbf{0}, \sigma^2 \mathbf{I}_n\right)$ and $p(\mathbf{x}|\mathbf{z}) = \mathcal{N}\left(\mathbf{f}_\theta^{-1}(\mathbf{z}), \sigma^2 \mathbf{I}_n\right)$. Plugging in these choices in the lower bound gives:

$$\log p(\mathbf{x}) \geq n(\log(2\pi\sigma) + \frac{1}{2})$$

$$+ \mathbb{E}_{\boldsymbol{\epsilon}} \left[ \log p(\mathbf{z}) - n\log(2\pi\sigma) - \frac{1}{2\sigma^2}(\mathbf{x} - \mathbf{f}_\theta^{-1}(\mathbf{z}))^T(\mathbf{x} - \mathbf{f}_\theta^{-1}(\mathbf{z})) + \log \left| \det \left( \frac{\partial \mathbf{f}_\theta(\tilde{\mathbf{x}})}{\partial \tilde{\mathbf{x}}} \right) \right| \right]$$

$$= \frac{n}{2} + \mathbb{E}_{\boldsymbol{\epsilon}} \left[ \log p(\mathbf{z}) - \frac{1}{2\sigma^2}\boldsymbol{\epsilon}^T\boldsymbol{\epsilon} + \log \left| \det \left( \frac{\partial \mathbf{f}_\theta(\tilde{\mathbf{x}})}{\partial \tilde{\mathbf{x}}} \right) \right| \right]$$

$$= \mathbb{E}_{\boldsymbol{\epsilon}} \left[ \log p(\mathbf{z}) + \log \left| \det \left( \frac{\partial \mathbf{f}_\theta(\tilde{\mathbf{x}})}{\partial \tilde{\mathbf{x}}} \right) \right| \right].$$

If we now let the noise level go to zero, $\sigma \to 0$, we will get that $\tilde{\mathbf{x}} \to \mathbf{x}$, $\mathbf{z} \to \mathbf{f}_\theta(\mathbf{x})$, which leaves us with

$$\log p(\mathbf{x}) = \log p(\mathbf{z}) + \log \left| \det \left( \frac{\partial \mathbf{f}_\theta(\mathbf{x})}{\partial \mathbf{x}} \right) \right|,$$

where the bound thus becomes tight as the noise level is decreased to zero. This is the usual log likelihood for normalising flow models.

## B  VDAE APPROXIMATE POSTERIOR FOR SMALL NOISE LEVEL

For our VDAE model, as specified in Section 2, we defined $\tilde{\mathbf{x}} = \mathbf{x} + \boldsymbol{\epsilon}$ and $\mathbf{z} = \mathbf{f}_\theta(\tilde{\mathbf{x}})$. Our sampled latents $\mathbf{z}(\tilde{\mathbf{x}}) = \mathbf{z}(\mathbf{x}, \boldsymbol{\epsilon})$ are thus a function of both the original uncorrupted data $\mathbf{x}$ as well as the added noise $\boldsymbol{\epsilon}$. To gain further insight into the approximate posterior distribution $q_\theta(\mathbf{z}|\mathbf{x})$ this implies, we perform a Taylor expansion of $\mathbf{z}(\mathbf{x}, \boldsymbol{\epsilon})$ around $\boldsymbol{\epsilon} = \mathbf{0}$ which gives

$$\mathbf{z} = \mathbf{f}_\theta(\mathbf{x}) + \mathbf{J}\boldsymbol{\epsilon} + \mathcal{O}(\boldsymbol{\epsilon}^2),$$

where $\mathbf{J} \equiv \frac{\partial \mathbf{f}_\theta(\mathbf{x})}{\partial \mathbf{x}}$ is the Jacobian matrix of $\mathbf{f}_\theta(\mathbf{x})$. If the scale of the noise becomes very small, the $\mathcal{O}(\boldsymbol{\epsilon}^2)$ term becomes negligible and we thus have $\mathbf{z} \approx \mathbf{f}_\theta(\mathbf{x}) + \mathbf{J}\boldsymbol{\epsilon}$. If we further specify the noise to be Gaussian, $q(\boldsymbol{\epsilon}) = \mathcal{N}\left(\mathbf{0}, \sigma^2 \mathbf{I}_n\right)$, like we did in the last section to show equivalence with normalising flows, we then have that

$$\lim_{\sigma \to 0} q_\theta(\mathbf{z}|\mathbf{x}) = \mathcal{N}\left(\mathbf{f}_\theta(\mathbf{x}), \sigma^2 \mathbf{J}\mathbf{J}^T\right).$$

Using this form of approximate posterior in combination with $p(\mathbf{x}|\mathbf{z}) = \mathcal{N}\left(\mathbf{f}_\theta^{-1}(\mathbf{z}), \sigma^2 \mathbf{I}_n\right)$, as in the last section, then makes our variational lower bound equivalent to the standard log-likelihood for normalising flows.

In order to better understand the relationship between normalising flows and VAEs we experiment with using $q_\theta(\mathbf{z}|\mathbf{x}) = \mathcal{N}\left(\mathbf{f}_\theta(\mathbf{x}), \sigma^2 \mathbf{J}\mathbf{J}^T\right)$ as our approximate posterior in various models. Here we also make use of the freedom the VAE perspective gives us to reduce the dimension of $\mathbf{z}$ as compared to the data $\mathbf{x}$.

## C  MODEL INITIALISATION

**Residual blocks** are initialised to be invertible by reducing them to identity mappings. For each residual block $\mathbf{y} = \mathbf{x} + \texttt{ReLU}\left(\texttt{ReLU}\left(\dots \texttt{ReLU}\left(\mathbf{x}\mathbf{W}_1\right)\dots\right)\mathbf{W}_{L-1}\right)\mathbf{W}_L$ with $\mathbf{x}, \mathbf{y} \in \mathbb{R}^{1 \times n}$ and

$\mathbf{W}_i \in \mathbb{R}^{n \times n}$ we zero-out all elements of $\mathbf{W}_L$. The same scheme was employed for initialising additive coupling blocks, which can be viewed as residual blocks of a restricted form.

**Projection layers** reduce dimensionality of their inputs using a linear map $\mathbf{y} = \mathbf{x}\mathbf{W}$ with $\mathbf{x} \in \mathbb{R}^{1 \times n}$, $\mathbf{y} \in \mathbb{R}^{1 \times m}$ and $\mathbf{W} \in \mathbb{R}^{n \times m}$. This generally leads to loss of information and makes model training harder. To mitigate this effect we initialise the rows of $\mathbf{W}$ using a set of $m$ random orthogonal vectors. The decoder projection layers, mapping data to higher dimensions, are then initialised to $\mathbf{W}^T$.

## D  HYPER-PARAMETERS AND ARCHITECTURES

All models were trained for 1000 epochs using the ADAM optimiser (Kingma & Ba, 2014) with a batch size of 1000 samples. To improve stability of the training, the learning rate was warmed up from $10^{-5}$ to the chosen learning rate (see below) over the first 10 epochs. Further, the KL term was warmed up by linearly annealing its weight $\beta$ from 0 to 1 over the first 100 epochs (Bowman et al., 2015).

For each experiment, the learning rate schedule $S \in \{\text{linear, none}\}$, learning rate $\alpha \in \left(10^{-5}, 10^{-3}\right)$ and ADAM optimiser parameters $\beta_2 \in \{0.9, 0.99, 0.999, 0.9999\}$ and $\epsilon \in \left\{10^{-4}, 10^{-5}, 10^{-8}\right\}$ were determined by using Bayesian optimisation of the ELBO on the validation set.

**NICE**  When implementing the model (standalone, or part of VDAE), we closely followed the architecture and hyper-parameters described in Dinh et al. (2014). Namely, the network consisted of 4 additive coupling blocks, each with 5 fully-connected hidden layers of size 1024 with `ReLU` activations, followed by a linear layer (see Appendix C). Dimension partitioning was alternated between even and odd dimensions after every block. When used as a standalone model, a $L_2$ regularisation with weight $\lambda = 0.01$ was used to improve sample quality.

**L-VDAE and vanilla VAE**  *When not used in conjunction with a NICE model in the encoder*, the L-VDAE and VAE models employed a fully-connected ResNet architecture with $B$ consecutive residual blocks followed by a linear projection layer to higher or lower dimensions. In the encoder, the last projection layer parameterised the means of the Gaussian variational posterior (and, in case of VAE, a parallel projection layer parameterised the log-variances). A sequence of 4 residual-projection "blocks" was used with the last block $i = 4$ projecting to $m$ dimensions (dimensionality of the latents) and the blocks before it, respectively to $\min\left[2^i \cdot m, 28 \times 28\right]$ dimensions. Each residual block consisted of 2 hidden layers with `ReLU` activations followed by a linear layer (see Appendix C). The residual block hidden size $H \in \{32, 64, 128, 256, 1024\}$ and the block multiplicity $B \in \{1, 2, 3\}$ were chosen through Bayesian optimisation as described above.

Unless otherwise specified, *when used together with a NICE model*, the VDAE and L-VDAE models employed a ResNet architecture in the decoder. In this case, the ResNet architecture was chosen to closely resemble that of the NICE model. Specifically, hyper-parameter values $B = 1$ and $H = 1024$ were used, and no projection layers were employed.

**Priors**  We employed a logistic prior with $s = 1$ and $\mu = 0$ (as in Dinh et al. (2014)) for models that made use of the NICE flow (even if it was only used in the encoder network); and a factorised normal prior otherwise.

## E  ADDITIONAL SAMPLES

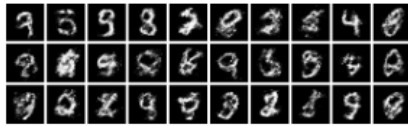

Figure 3: Samples from the VDAE model trained on the **continuous** MNIST data with different *learned* noise levels $\log \sigma$. Provided as a reference for samples obtained from models with fixed noise levels (Figure 4).

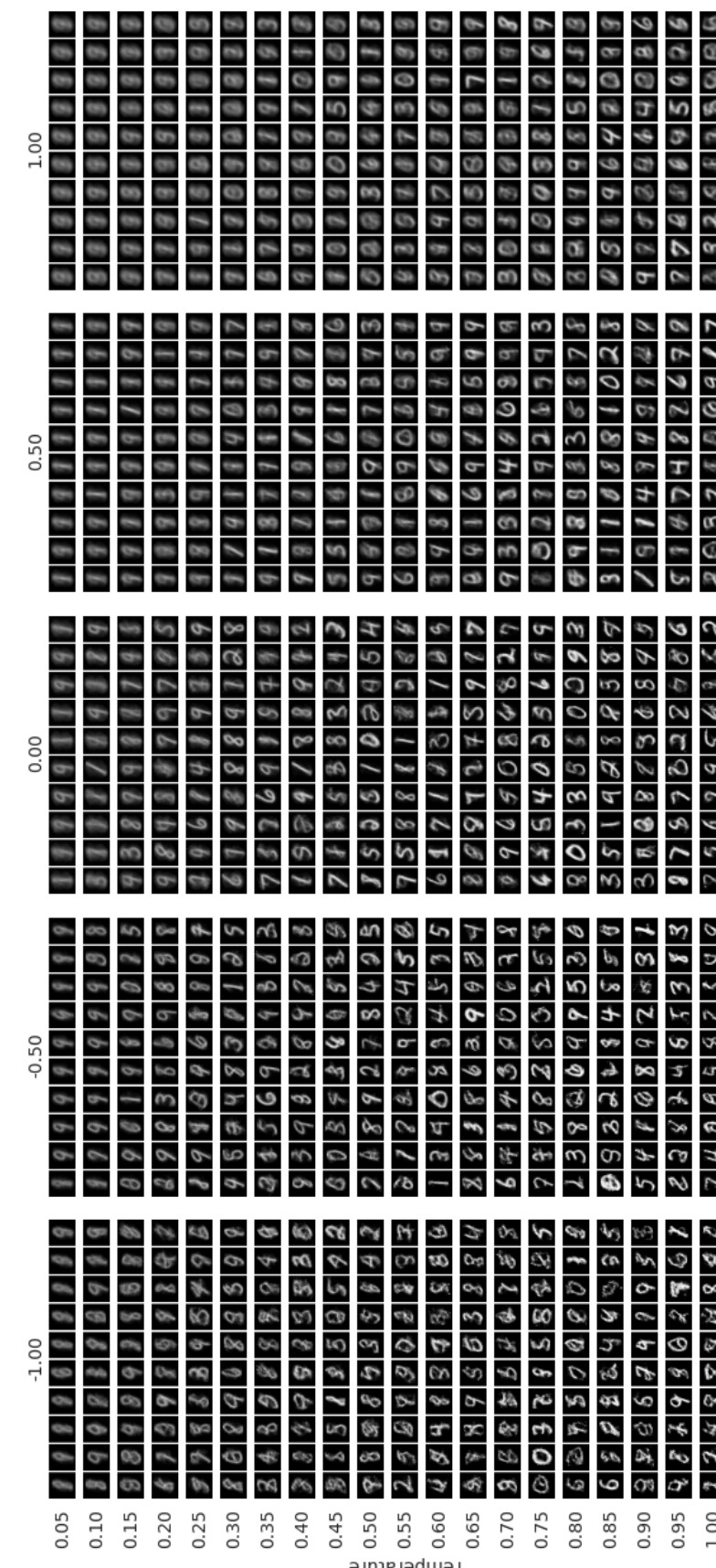

Figure 4: Samples from the VDAE model trained on the **continuous** MNIST data with different *fixed* noise levels $\log \sigma$. Sample quality deteriorates at the extremes of the noise level spectrum: at high noise levels the model appears to be unable to learn the distribution, whereas at low noise levels the model appears to focus too much on the reconstruction error instead of organising the latent space. Just as in DAEs, the noise variance $\sigma^2$ in VDAEs can be used as a regulariser; and can be tuned for sample quality.

