# OpenReview forum: "On the relationship between Normalising Flows and Variational- and Denoising Autoencoders"
_ICLR.cc/2019/Workshop/DeepGenStruct — DeepGenStruct 2019_

### Official Review · AnonReviewer1 · 2019-04-16

**Rating:** 3
**Confidence:** 2

**Review:**

This paper applies normalizing flows into denoising autoencoders, and derives a variational lower bound when the posterior has this form. Experiments on MNIST show improvements over VAE, but worse than other NF models.

pros: The paper is well written and easy to follow. It does a good job in reviewing related work.

cons: While it combines VAE, NF and DAE, no particular novel technique is introduced, and it‘s no better than existing models, so the significance of the proposed framework is unclear. In addition, the expensive complexity of L-VDAE makes it  difficult to scale to high-dimensional data. Nevertheless, the topic is very relevant and I think it's worth discussing at this workshop.

---

### Official Review · AnonReviewer2 · 2019-04-16
**Interesting exposition of unifying NFs, VAEs, and DAEs**

**Rating:** 3
**Confidence:** 3

**Review:**

This paper proposes a model family that unifies NFs, VAEs and DAEs. It also introduces an extension of this model that allows for using non-invertible encoders (e.g. projection to a smaller dimensionality) and discrete data. Overall, the idea is promising, but the empirical results are not strong enough to warrant a strong commendation.

Pros
- The proposed model that blends NFs, VAEs and DAEs is original, and generalises over standard NFs in that it allows non-zero noise levels.
- When latent dimensionality is smaller than the input space, the model consistently outperforms the VAE baseline.

Cons
- Performance deteriorates for bigger latent dimensionalities (e.g. when n=m)

---

### Decision · Program_Chairs · 2019-04-19
**Acceptance Decision**

**Decision:**

Accept

**Comment:**

Accepted